# Modelling and classifying joint trajectories of self-reported mood and pain in a large cohort study

**Rajenki Das** [1]*, **Mark Muldoon**[1], **Mark Lunt**[2], **John McBeth**[2], **Belay Birlie Yimer**[2], **Thomas House**[1]

**1** Department of Mathematics, University of Manchester, Manchester, United Kingdom, **2** Centre for Epidemiology Versus Arthritis, University of Manchester, Manchester, United Kingdom

* rajenki.das@manchester.ac.uk

## Abstract

It is well-known that mood and pain interact with each other, however individual-level variability in this relationship has been less well quantified than overall associations between low mood and pain. Here, we leverage the possibilities presented by mobile health data, in particular the "Cloudy with a Chance of Pain" study, which collected longitudinal data from the residents of the UK with chronic pain conditions. Participants used an App to record self-reported measures of factors including mood, pain and sleep quality. The richness of these data allows us to perform model-based clustering of the data as a mixture of Markov processes. Through this analysis we discover four endotypes with distinct patterns of co-evolution of mood and pain over time. The differences between endotypes are sufficiently large to play a role in clinical hypothesis generation for personalised treatments of comorbid pain and low mood.

## Author summary

Mood and pain are known to interact, and a mobile-phone application recorded information on the variations of mood and pain amongst people in the UK. Using this data, we observed that people have a general tendency of feeling the same mood and pain the next day. Studying further, we were able to separate the people into four groups- three of which were quite different from the general pattern of mood pain. The additional patterns we saw were 1) their mood and pain deteriorating the next day, 2) their mood and pain improving the next day and 3) mood is improving but pain deteriorates the next day. These additional characteristics tell us that there is no definite way that mood and pain are associated for everyone, and personalised treatment to tackle challenges in mood and pain can deliver better results.

## Introduction

Mental disorder has been associated with a substantial excess in all-cause mortality risk [1]. It is often accompanied by mood disorders which, according to the World Health Organisation

**Data Availability Statement:** Our work involved secondary analysis of data from the project Cloudy with a Chance of Pain (see public website at https://www.cloudywithachanceofpain.com). The data is

in the process of being made available to researchers outside the University of Manchester via a trusted research environment. Please contact Professor Will Dixon [will.dixon@manchester.ac.uk] to enquire about access.

**Funding:** RD and TH are supported by the Engineering and Physical Sciences Research Council. ML, JMcB and BBY are supported by Centre for Epidemiology Versus Arthritis. TH is also supported by the Royal Society, the Medical Research Council and the Alan Turing Institute for Data Science and Artificial Intelligence. The funders had no role in study design, data collection and analysis, decision to publish, or preparation of the manuscript.

**Competing interests:** The authors have declared that no competing interests exist.

(WHO) [2], are one of the leading causes of disability. Mental health can suffer due to many social, physical and other factors, and mathematical approaches are uniquely placed to disentangle these complex issues. In view of the difficulty in clearly defining "mental illness" itself, simply linking its absence with positive mental health is not enough [3, 4]. One may not suffer from any "mental illness", yet not be mentally fit. So, identifying markers of mental health disorders remains a vital challenge.

Chronic pain is a persistent or intermittent pain that lasts for more than 3 months [5], and approximately one fifth of the population in the USA and Europe are affected by it [6]. Chronic pain can cause a lot of emotional distress and affect lifestyle by interrupting activities [7] thereby it can potentially lower a person's mood. Low mood and low self esteem often give birth to mental disorders like depression. [8] and [9] noted that depression is a frequent accompaniment to chronic pain while, [10] observed that those who suffer from depression often complain of pain. Depression, which is commonly associated with chronic pain [11, 12], is one of the leading contributors to global disease burden [13, 14]. It has been seen that chronic pain and depression tend to coexist [15] and the relationship between the two is widely studied. [16] showed that when a depressed mood was induced in patients with chronic back pain, their pain ratings increased, while participants with a happy mood had lower pain ratings. [11] observed evidence against the hypothesis of depression preceding the development of pain and indicated that pain may play a causal role for depression. Chronic pain could be due to presence of inflammatory diseases [17], which cause inflammation in the body that can produce cytokines which can lower mood [18], and according to [19], higher levels of biomarkers associated with inflammation are linked with depression. So today, the causal relationship of these associations between inflammation and mood disorders is said to be bidirectional [20–22].

It is widely recognised that healthcare increasingly involves dealing with comorbidities [23], and also personalisation of treatment plans [24]. Health issues such as mood disorders and conditions associated with chronic pain are often comorbid [25, 26], but the manner in which these conditions influence each other varying from person to person is still considerably uncertain. Mood disorders or depression can be treated in three ways: antidepressants, psychotherapy and electro-convulsive therapy (ECT) [27]. Chronic pain treatments can be based on multiple aspects of pain experience like the intensity and quality of pain, and use of rescue analgesic medications [28]. For certain types of chronic pain, drug therapy including intake of analgesics like non-steroidal anti-inflammatory drugs (NSAIDs) could be the option, while for others, a multimodal approach may be required [29] eg: a pharmacotherapy consisting analgesics and Cognitive–behavioural therapy (CBT) together can be effective when chronic pain and anxiety disorders co-occur [30]. But when dealing with both mood disorders and chronic pain, especially when considering only pharmaceutical interventions, it must be noted that the combined usage of anti-depressants and NSAIDs can have negative effect, as shown in [31, 32] where there observed a risk of intracranial haemorrhage although there was no such association found in independent use.

Nowadays, technology is making its presence felt in several sectors, one of which is the health sector. It is only in the early 21$^{st}$ century that eHealth, a broad term for the combined usage of electronic and communication technologies in the health sector, emerged [33]. Many novel ways have developed to tackle healthcare issues and provide support. From wearable accessories to smartphone applications, all of these are aiding healthcare. From a global perspective, e-health is useful in dissemination of health information as well as ensuring that the most updated information is used to improve the health [34, 35]. The WHO's Global Observatory for eHealth defines mobile health (mHealth) as "medical and public health practice supported by mobile devices, such as mobile phones, patient monitoring devices, personal digital

assistants (PDAs), and other wireless devices". mHealth is a powerful way to cater to individual requirements. Few of the benefits of the mHealth tools, especially for the purpose of research, are: (i) cost-effectiveness while collecting voluminous amount of data; (ii) more honesty in answers received as there is no direct human intervention in collection of data; and (iii) convenience of easily linking mHealth apps to other link to other sensing tools. More than one in four people are affected by mental health disorders like depression, anxiety etc. worldwide [36], and digital technology interventions show the potential to extend support to those who suffer from mental health problems. There is a growing need to make digital based mental health care aid accessible to as many people as possible [37], and in this study, we make use of digital health data to analyse mood-pain patterns in a cohort of residents of the UK with chronic pain conditions.

We explore the association between pain and mood by analysing long records of self-reported, daily data collected using a mobile phone application. We perform clustering on the basis of the transitions of mood-pain and show how an intervention to improve low mood or high pain symptoms can affect the clusters differently.

## Methods

### Data

We use data from the Cloudy with a Chance of Pain study [38, 39], which was conducted to investigate the relationship between weather and pain, but in doing so created an extremely rich dataset suitable to answer a diversity of research questions. Data were collected from January 2016 to April 2017 from participants resident in the UK who were aged 17 or above and had experienced chronic pain for at least 3 months preceding the survey [40]. These participants were recruited from the general public, with particular support from the charity Versus Arthritis (then Arthritis Research UK) rather than via the healthcare system, meaning that it is important to interpret these results as applying to people's experiences of chronic pain in the wider community rather than purely under clinical management.

The cohort had 10,584 survey participants, each of whom was asked to rate their symptoms and other variables on a mobile application in five ordinal categories (e.g. pain scores ranged from 1 for no pain to 5 for very severe pain). Data were recorded for pain interference, sleep quality, time spent outside, tiredness, activity, mood, well-being, pain severity, fatigue severity and stiffness on a daily basis. However, participants did not always report all the data daily so we considered only those (Mood, Pain) states where both the values are available, leaving us with $N = 9990$ participants for our analysis. The average (rounded off) length of trajectories is 44 observations.

In this paper we analyse trajectories of pairs of self-reported pain severity and mood scores. Participants were asked to provide information on these on a five-point Likert scale, with accompanying text for each of the ordinal levels. For mood, a score of 1 represents worst mood and 5 represents best, whereas for pain a score of 1 represents least pain and 5 represents most.

For easier analysis of the data and interpretation of results, we regrouped the severity of mood and pain into two categories each on the basis of the descriptions associated with each ordinal value. Mood scores of 1–3 and 4–5 were labelled Bad (B) and Good (G) respectively, while pain levels of 1–2 and 3–5 were, respectively, labelled Low (L) and High (H). Thus, at a given time, a participant's mood and pain scores fall into one of four states: GL; GH; BL; and BH. Full details are shown in Table 1.

Participants self-reported diagnoses, and also provided information on age, sex, pain condition diagnosed and the site of pain. They might have more than one condition and site of pain.

**Table 1. Mood and pain scores, descriptions, and binary classifications.** *Score* is the value on a Likert scale available to participants, *Text* is the description presented to them when recording these data, and *Binary* is our binary classification into 'Good' (G) or 'Bad' (B) for Mood, and 'Low' (L) and 'High' (H) for Pain.

| Mood | | | Pain | | |
|---|---|---|---|---|---|
| *Score* | *Text* | *Binary* | *Score* | *Text* | *Binary* |
| 1 | Depressed | Bad | 5 | Very severe pain | High |
| 2 | Feeling low | Bad | 4 | Severe pain | High |
| 3 | Not very happy | Bad | 3 | Moderate pain | High |
| 4 | Quite happy | Good | 2 | Low pain | Low |
| 5 | Very happy | Good | 1 | No pain | Low |

The list of conditions includes Rheumatoid arthritis, Osteoarthritis, Spondyloarthropathy, Gout, Unspecific arthritis, Fibromyalgia, Chronic headache and Neuropathic pain. The list of sites of pain taken in this analysis includes mouth or jaw, neck or shoulder, back pain, stomach or abdominal, hip pain, knee pain, and hands.

Code for this study is made available at: https://github.com/rajenkidas/EM-clustering-on-Markov-Chains. The data is scheduled to be made available to the wider research community via a trusted research environment in 2023.

### Residual analysis

We performed an initial data analysis based on Pearson residuals, looking for notable patterns in the co-evolution of mood and pain over time using standard methodology as outlined by e.g. [41]. Such an analysis particularly helps to visualise the ways in which observed patterns deviate from a simple 'null' model.

We begin by visualising a matrix of transitions observed in the data. Let $Y$ be the count matrix whose element $Y_{ij}$ denotes the total number of observed transitions—across all participants—from state $i$ one reporting day to state $j$ the next reporting day. We then perform Pearson residual analysis to compare observed transition probabilities with the expected values given a specified 'null' model assumption, which we fit by maximum likelihood estimation. Throughout this work we will use the standard result that the maximum likelihood estimator for a probability of an outcome is the observed number of such outcomes divided by the number of observations under binomial and Poisson sampling (which we also assume throughout as appropriate).

We have seen that participants are most likely to remain in their current state rather than move to another one. That is, their mood and pain scores do not usually change from one day to the next, as shown in Fig 1A. These observations allow us to define a simple first model for their behaviour and perform residual analyses as described below. In this exploratory analysis we work with the original data, so there are $n = 5 \times 5 = 25$ states.

We therefore define a null model in which the number of participants starting in state $i$ is $N_i$, the probability of staying in state $i$ is $\pi_i$ and when a person does change state, the probabilities $P_{ij}$ of a transition from state $i$ to state $j \neq i$ are uniform. The model parameters can then have maximum likelihood estimators (indicated with hats) as follows. For $i, j \in \{1, 2, \ldots, n\}$,

$$\hat{N}_i = \sum_{k=1}^{n} Y_{ik}, \quad \hat{\pi}_i = \frac{Y_{ii}}{\sum_{k=1}^{n} Y_{ik}}, \quad \hat{P}_{ij} = \begin{cases} \hat{\pi}_i & \text{if } i = j, \\ \dfrac{1 - \hat{\pi}_i}{n-1} & \text{otherwise,} \end{cases} \quad E_{ij} = \hat{N}_i \hat{P}_{ij}, \quad (1)$$

where $E_{ij}$ is the $(i, j)$-th element of the matrix of expected counts, **E**. The associated entry in the

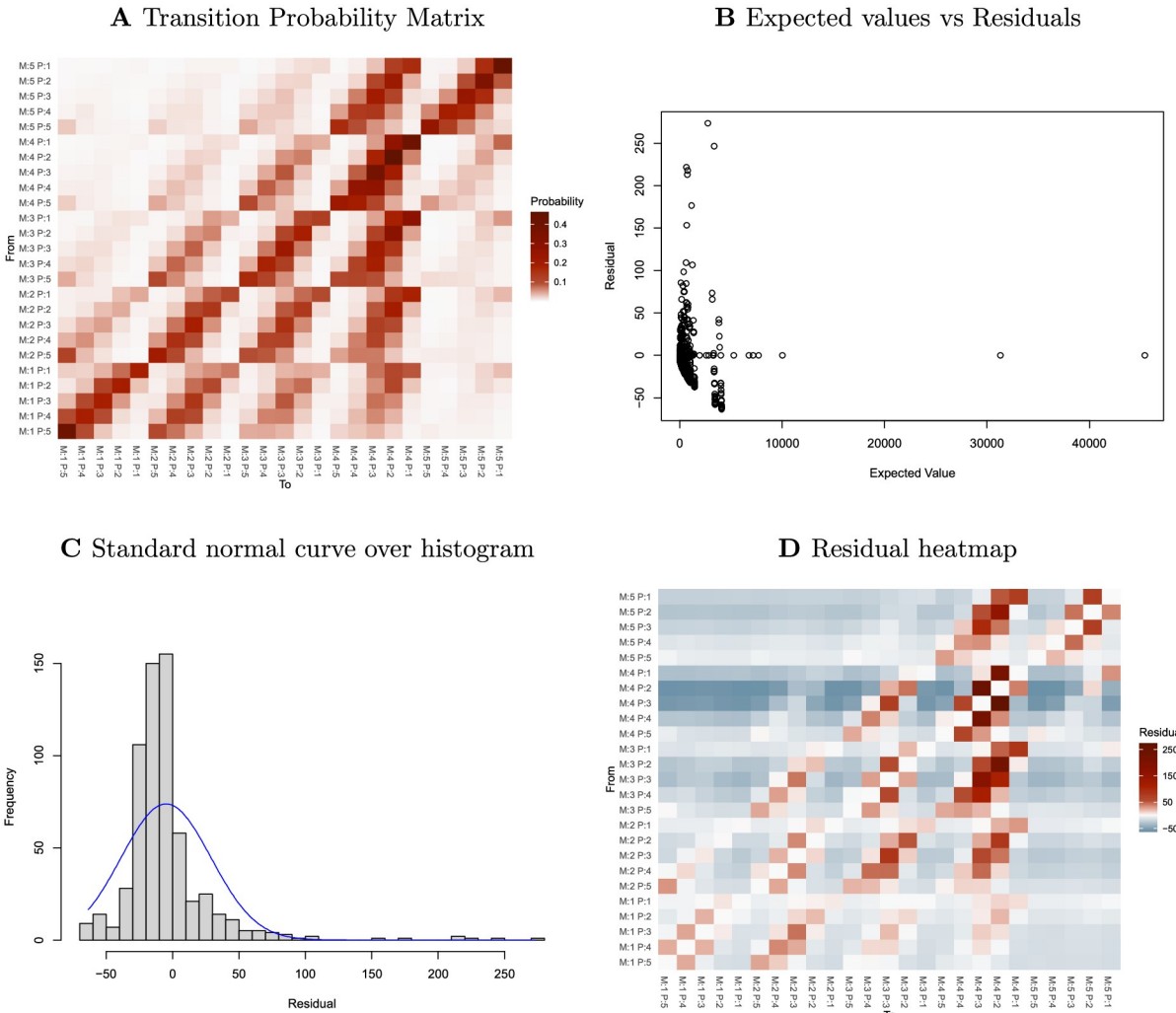

**Fig 1. A** is the heatmap of probabilities of transitions from one state to another. **B** is the scatter plot of expected values and the residuals. **C** shows a histogram of the residuals as well as a blue curve giving the probability density function of a normal distribution having the same mean and variance as the residuals. **D** is a heatmap of the matrix of residuals based on the model specified by Eq (1).

Pearson residual matrix **R** is then given by

$$R_{ij} = \frac{Y_{ij} - E_{ij}}{\sqrt{E_{ij}}}. \tag{2}$$

Since we expect such residuals to be asymptotically standard normal under the null [41], we will interpret these as values over 2 indicating significantly more events than expected under the null, and values under −2 indicating significantly fewer.

## Clustering analysis

In this section we outline methods used to classify the participants using unsupervised learning, organising the participants into clusters on the basis of their sequences of reduced (Mood, Pain) states: GL, GH, BL, BH.

**Model setup.**   We assume the sequence of self-reported mood-pain states $X = (X_t; t \geq 0)$ is generated by a Markov chain:

$$\Pr(X_{t+1} = j \mid X_0 = k_0, ..., X_t = i) = \Pr(X_{t+1} = j \mid X_t = i) =: P_{ij}, \tag{3}$$

where the $P_{ij}$ are called the chain's *transition probabilities*.

Our data consists of trajectories of mood-pain pairs that we reduce to matrices tabulating numbers of transitions observed for each participant individually. We then cluster these count-matrices by using the EM algorithm to fit a mixture of Markov chains with a distinct matrix of transition probabilities for each component of the cluster.

Let the number of states be $n$ and the number of participants be $S$. We write **C** for the matrix of total count of transitions from one state to another, and use $\mathbf{C}_s$ for the matrix of counts of transitions that appear in the trajectory of states of mood-pain of participant $s$. We note that **C** is distinguished from the count matrix **Y** introduced before in Residual Analysis section since now it involves only the four reduced states.

**The expectation-maximisation algorithm.**   The classical Expectation-Maximisation (EM) algorithm [42] provides a way to do maximum-likelihood estimation of parameters in a setting where some variables are unobserved or unknown. In our case, the latent variables are the classes to which the participants belong. The algorithm involves iteration of two alternating steps: the E, or *expectation* step, during which one computes the expected value of the log likelihood for the observed data, given the current estimates of the parameters, and the M, or *maximisation*, step during which one re-estimates the parameters is maximising the expected value as calculated in the E-step.

The details of this algorithm are given in S 1.3 of S1 Text. Its outputs are a number of clusters $K$, and an $S \times K$ matrix $\mathbf{\Gamma}$ such that its $(s, c)$-th element $\Gamma_{sc}$ is the probability that participant $s$ belongs to cluster $c$. Finally, cluster assignments are then made on the basis of the class membership probabilities: participants are assigned to whichever cluster they have the highest probability of belonging to.

**Associated stationary distribution.**   The stationary distribution for a Markov chain with $n \times n$ transition matrix **M** has probability $x_i$ associated with state $i$, where $\mathbf{x} = (x_i)$ solves the left Eigenvalue equation

$$x_k = \sum_{i=1}^{n} x_i M_{ik}, \tag{4}$$

where $k \in \{1, ..., n\}$, and we impose conditions ensuring that **x** is a probability vector: $x_i \geq 0$ and $\sum_{i=1}^{n} x_i = 1$.

The solution to Eq (4) need not be unique, but as the transition matrices of our problem are regular, we do get a unique stationary distribution for each component of the mixture [43]. That is, for each cluster, we get a distribution over the states BH, BL, GH and GL. Further, as the Markov chains are ergodic, the modelled expected fraction of time an individual participant spends in state $k$ is given by $x_k$.

## Intervention

In this section, we explore the prospect of alleviating low mood or high pain, which can be done by taking the appropriate treatment targeting mood or pain. We naïvely examine how the interventions could work by altering the transition probabilities associated with the clusters and see what effect this has on the cluster's stationary distribution. Throughout, we will let the

transition probability matrix before intervention be represented as:

$$
\begin{array}{c}
\\ BH \\ \\ BL \\ \\ GH \\ \\ GL
\end{array}
\begin{array}{cccc}
GL & GH & BL & BH \\
\left[\begin{array}{cccc}
M_{11} & M_{12} & M_{13} & M_{14} \\
M_{21} & M_{22} & M_{23} & M_{24} \\
M_{31} & M_{32} & M_{33} & M_{34} \\
M_{41} & M_{42} & M_{43} & M_{44}
\end{array}\right].
\end{array}
\tag{5}
$$

**Improving mood.** To model an improvement in mood, we increase the probabilities of transitions from states of bad mood to those with good mood. We get an updated transition matrix $\mathbf{M}'_c$ for every cluster $c$ in the following way:

$$
\begin{array}{c}
BH \\ \\ BL \\ \\ GH \\ \\ GL
\end{array}
\begin{array}{cccc}
GL & GH & BL & BH \\
\left[\begin{array}{cccc}
M_{11} + \beta_M & M_{12} + \beta_M & 0.8 \times (M_{13} + M_{14} - 2\beta_M) & 0.2 \times (M_{13} + M_{14} - 2\beta_M) \\
M_{21} + \beta_M & M_{22} + \beta_M & 0.8 \times (M_{23} + M_{24} - 2\beta_M) & 0.2 \times (M_{23} + M_{24} - 2\beta_M) \\
M_{31} & M_{32} & M_{33} & M_{34} \\
M_{41} & M_{42} & M_{43} & M_{44}
\end{array}\right],
\end{array}
\tag{6}
$$

where the rows are labelled by the (Mood, Pain) states from which the transition starts, while the columns are labelled by the states to which it goes. Here $\beta_M$ must be chosen so that all transition probabilities remain in the range $0 \le M'_{cij} \le 1$. For our fitted transition matrices, these constraints mean that $0 \le \beta_M \le 0.15$.

One can see that we distribute the probabilities disproportionately between transitions to BH and BL from BH and BL. This has been done to reduce the probability of moving to BL, which we wish to model as less likely under an intervention assumed to be beneficial. In fact, in general the probability of moving to good mood from bad mood could have been achieved in numerous other ways through changes to the full matrices. The choice used here permits a more substantial increase in the probabilities of improved mood than simpler formulæ, many of which are strongly constrained by the necessity of keeping all probabilities to the laws of probability.

**Improving pain.** Similar to improvement of mood, we considered altering the transition probabilities to improve pain, which means increasing probability of transitioning to low pain through adding and subtracting $\beta_P$ as shown below for the updated transition probability matrix $\mathbf{M}'_c$ for every cluster $c$ in the following way:

$$
\begin{array}{c}
BH \\ \\ BL \\ \\ GH \\ \\ GL
\end{array}
\begin{array}{cccc}
GL & GH & BL & BH \\
\left[\begin{array}{cccc}
M_{11} + \beta_P & 0.8 \times (M_{12} + M_{14} - 2\beta_P) & M_{13} + \beta_P & 0.2 \times (M_{12} + M_{14} - 2\beta_M) \\
M_{21} & M_{22} & M_{23} & M_{24} \\
M_{31} + \beta_P & 0.8 \times (M_{32} + M_{34} - 2\beta_M) & M_{33} + \beta_P & 0.8 \times (M_{32} + M_{34} - 2\beta_M) \\
M_{41} & M_{42} & M_{43} & M_{44}
\end{array}\right],
\end{array}
\tag{7}
$$

Here $0 \le \beta_P \le 0.2$. In both cases, we then examine the resulting changes in the stationary distributions to see the consequences of the intervention for each cluster individually.

### Ethics statement

This is a secondary analysis. Ethical approval of the primary data was obtained from the University of Manchester Research Ethics Committee (ref: ethics/15522) and from the NHS IRAS (ref: 23/NW/0716).

## Results

### Residual analysis

The resulting transition probability matrix is illustrated in Fig 1A, which is a heatmap illustrating the probabilities with which participants switch from one pair of mood-pain scores to another. It is based on the original data and so has $5 \times 5 = 25$ possible states and $25 \times 25 = 625$ possible transitions. It has rows labelled by a current mood-pain pair and columns labelled by the mood-pain pair on the following day.

Note that the diagonal elements—those that correspond to remaining in the same state on successive days—have high probabilities. The entries at upper right and lower left, which correspond, respectively, to the worst and best mood-pain scores, are especially large (near their maximum value, 1) indicating that participants at the extremes of the scale have a strong tendency to remain there.

In Fig 1B and 1C which illustrate the distribution of residuals for this model as computed with Eq (2), clearly show that the residuals do not appear to be normally distributed. Looking at the residual heatmap in Fig 1D, we can say that the naïve model specified by Eq (1) does not describe the data well.

This suggests we try another model or check for latent variables or clusters. We try another model in the S 1.5 of S1 Text which showed an improvement in fitting since the residual range decreases in S8 Fig, but it still did not fit the data well as we see in S9 Fig. So we move on to clustering the data, as explained in the next section.

### Clustering

We found four clusters using the EM algorithm to do model-based clustering using a mixture of Markov chains, as illustrated in Fig 2, where the clusters are represented by heatmaps of their transition matrices.

Before describing the clusters, it should be noted that GL is the best state as both mood and pain are good, while BH is the least preferable state to be in as both mood and pain are bad here. Based on the transition probabilities, the four clusters for mood-pain dynamics can be broadly characterised as:

**Cluster 1**: Movement to the least preferable state. 1783 members.
  Here, we see that there are high probabilities of moving to the state where there is bad mood and high pain.

**Cluster 2**: Movement to the ideal state. 1558 members.
  In this cluster, we observe, irrespective of the current state, a participant is most likely to be in good mood and low pain the next day.

**Cluster 3**: Good mood, high pain. 2019 members.
  In this cluster, the dominant movement is to the state with good mood and high pain.

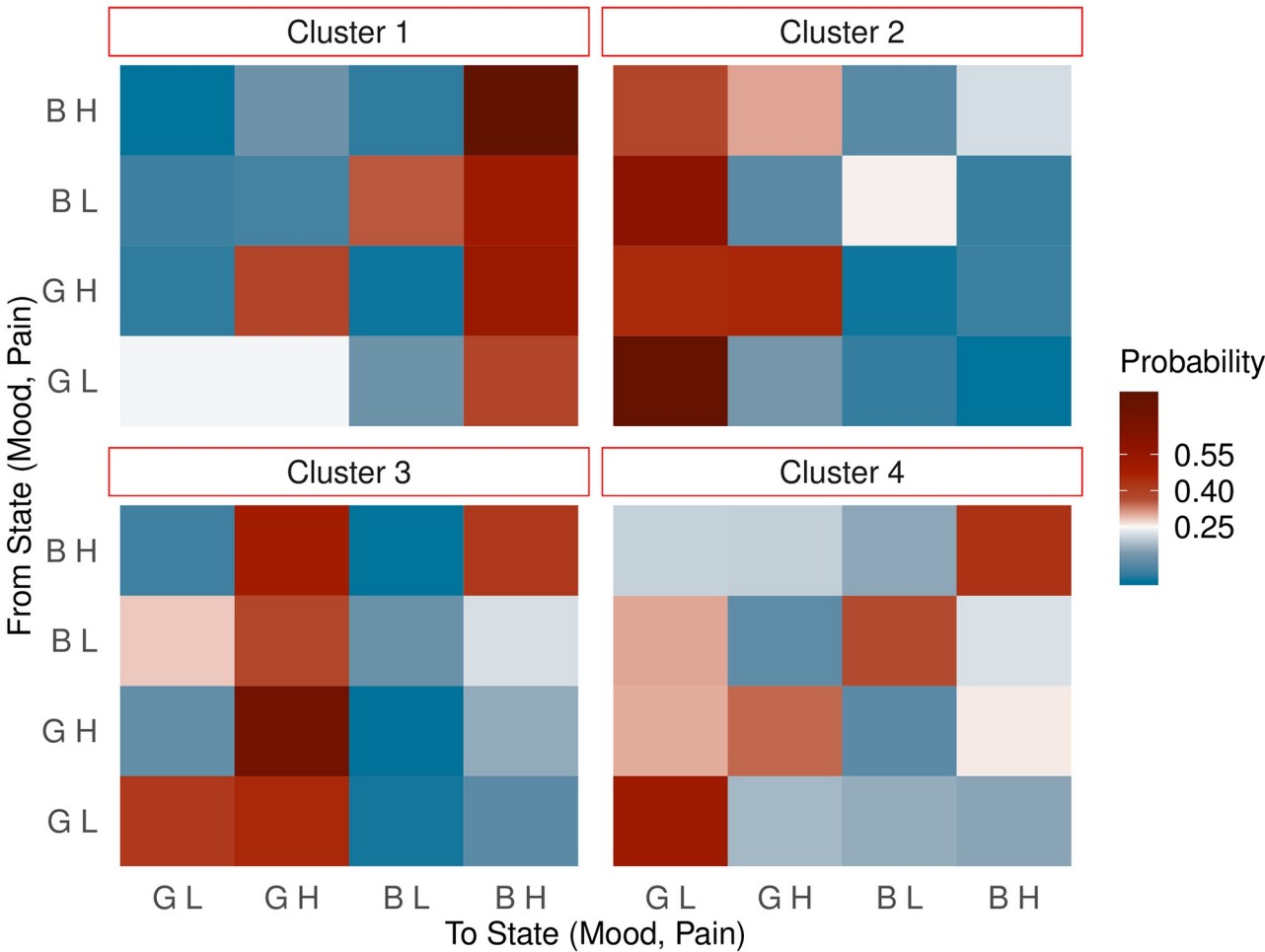

**Fig 2. Heatmaps of the transition probability matrices for the four clusters where G, B, L and H imply good mood, bad mood, low pain and high pain respectively.**

**Cluster 4**: Remain in the same state. 4630 members.

Most of the participants tend to stay in the same state.

Given the total of 9990 participants, we see that it is most common for participants (46%) to be members of Cluster 4 involving staying in the same state, which is consistent with our exploratory analysis of transitions. The smallest cluster (number 2) with 16% of participants, consists of those who tend to the ideal state, but at the same time, not many (18%) are in Cluster number 1 that tends to the worst state. The remainder (20%) belong to the third cluster: good mood, high pain.

In Fig 3, we present a set of comparisons of properties of the clusters. The stationary distributions as defined by Eq (4) are shown in Fig 3A, and as would be expected from the full estimated transition probability estimates they are derived from: Cluster 1 has most probability mass on BH; Cluster 2 has most probability mass on GL; Cluster 3 has most probability mass on GH; and Cluster 4 has evenly distributed probability masses.

In Fig 3B, we compare age distributions by sex and cluster, seeing that Clusters 1 and 4 have comparable age distributions, but Cluster 3 is associated with older ages than these two

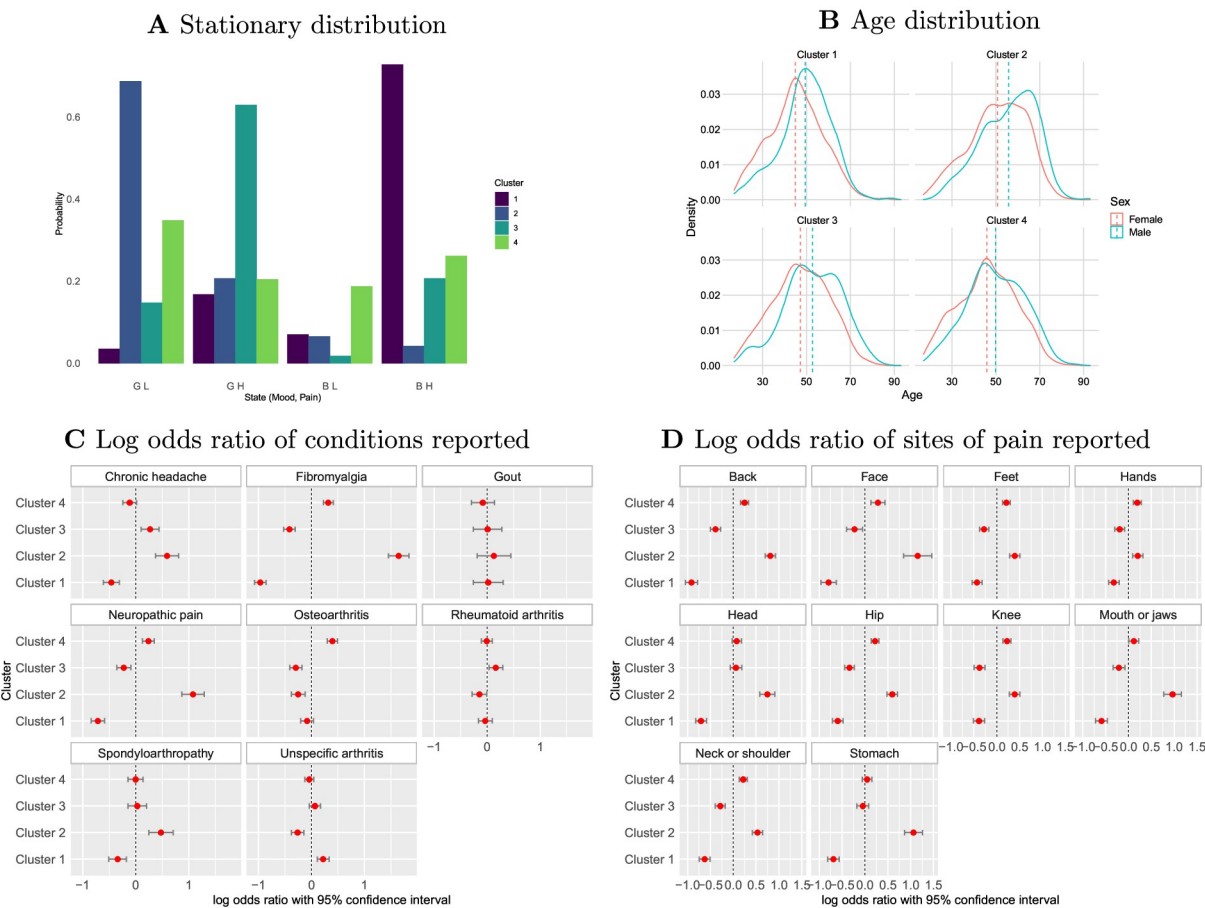

**Fig 3. A** stationary distributions for the four clusters. **B** age distributions for each cluster. **C** and **D** represent log odds ratio of condition and site of pain respectively, per cluster.

and Cluster 2 is associated with older ages than all three other clusters. Males are typically older than females in all clusters.

Participants had one or more conditions and sites of pain and the log odds ratios for these per cluster are shown in Fig 3C and 3D. These show that while some conditions and sites such as gout and hands are not strongly associated with any cluster, for others this is not the case. Fibromyalgia and stomach pain are particularly strongly associated with Cluster 2, for example.

## Intervention

We look at how interventions could work help alleviate the symptoms of bad mood and high pain.

In Fig 4A, Cluster 2 shows least improvement in mood, while Cluster 1 shows the most followed by Cluster 4. Decrease in state BH is the highest for Cluster 1, followed by Cluster 4 and least for Cluster 2. Overall, Cluster 1 shoes the most drastic changes in probability distribution while Cluster 2 is the least. We also note that in the case of improving mood from bad mood, state BL probability drops for Clusters 2 and 4, while it increases for 1 and 3.

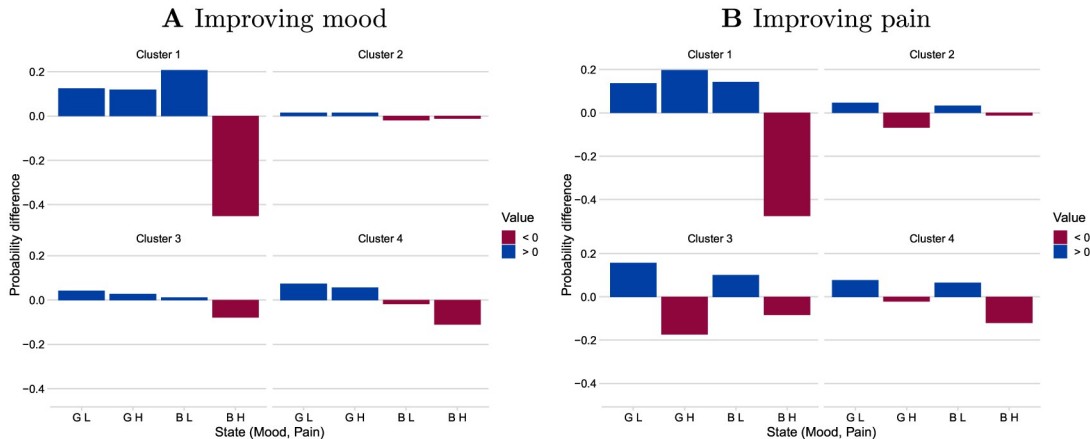

**Fig 4. Change in stationary distributions by an intervention.** In **A**, $\beta_M = 0.15$ is added to probabilities of transitioning from bad mood to good mood, while in **B**, $\beta_P = 0.15$ has been added to the probabilities of transitions from high pain to low pain.

In Fig 4**B**, we again find Cluster 1 with maximum changes. When intervened to improve pain by lessening the intensity of pain, probability of GH state improves only for Cluster 1.

## Discussion

In this work, we have performed an analysis of joint trajectories of mood and pain of participants in the large mobile health cohort, "Cloudy with a Chance of Pain". In addition to analysis of the full set of transitions using residuals, we performed clustering on transitions between a simplified set of variables and in doing so found four digital behavioural phenotypes on the basis of people's past trajectories of their mood-pain states. This suggests that even though mood and pain have been known to be correlated, the association may not be generalised in one single way for an entire population.

Previous studies on mood-pain relationships have tended to reach the conclusions on universal associations between mood and pain—i.e. generalising the result for everyone. The clusters found in this study emphasise that mood-pain relationships may differ between (groups of) individuals. The varying relationships between mood and pain, as shown by the clusters, highlights that such variability should be taken into account when considering expected future associations—for example, in a clinical prediction model, an approach of personalising forecasts could be taken.

Going beyond association to look at mechanism and causation, we stress that we have not performed causal inference and so results should all be interpreted as indicative of (potential) association magnitudes rather than as causal statements. Nevertheless, the interpretability of the observed clusters and their diversity in terms of e.g. conditions and sites of pain represented suggests that there may be associated endotypes—i.e. clusters representing distinct mechanisms of disease. If such causally distinct groups exist, then our hypothetical investigation of interventions that target either mood or pain individually suggests that we might expect clinically significant differences from different treatment depending on an individual's endotype.

Our study has some limitations that should be borne in mind when interpreting results. The first of these is, as discussed above, that we consider associations rather than causation. Furthermore, we have assumed missing values—primarily arising when participants did not enter data on one day—can be ignored and so have removed them; although this is not a

major component of the data an alternative would be to model non-response as a separate value. Along related lines, the simplification of the state space, while necessary for the EM algorithm to produce plausible transition matrices for each cluster, involves some information loss and this leaves open the possibility of more sophisticated methodology to perform the clustering. Also, factors common to all observational studies such as this one are important to bear in mind, particularly that individuals are selected from the general rather than a clinical population.

Extension of the work presented here could include applying the same methodology to more datasets to check if the phenotypes found are reproducible. This would further strengthen the likelihood of different causal relationships holding within clusters. To make a fuller assessment of likely causation, however, expected relationships between all observed and unobserved variables would need to be specified, and ideally intervention studies run. Additionally, this work can be extended by including socio-economic factors, extra latent variables like sleep quality, environment etc. Another direction would be to apply different techniques to this dataset, such as linear model based approaches that can identify latent classes [44, 45]. Different methods may allow the Markovian assumption made in our work to be relaxed, allowing for e.g. consideration of patterns in longer sequences of data, but at the cost of the ability to model out of sample behaviour as Markov chains allow.

Ultimately, our hope is that work on observational data such as that presented here can aid with hypothesis generation for future clinical studies of more personalised interventions for common problems such as low mood and chronic pain.

## Supporting information

**S1 Fig. Frequency of states.**
(TIF)

**S2 Fig. Overall age distribution, included for comparison with Fig 3B.**
(TIF)

**S3 Fig.** A and B indicate the proportion of participants in a cluster reporting, respectively, a given condition and site of pain. C and D show the proportions of participants with, respectively, a given condition or site of pain who fall into each cluster.
(TIF)

**S4 Fig. Negative Log-Likelihood as a function of the number of components.**
(TIF)

**S5 Fig. Difference of negative Log-Likelihoods of models with number of clusters $k + 1$ and $k$.**
(TIF)

**S6 Fig. Dotted lines represent negative Log Likelihood gradients extrapolated from the difference between clusters 1 and 2, and clusters 9 and 10.**
(TIF)

**S7 Fig. Bayesian Inference Criterion (BIC) per model.**
(TIF)

**S8 Fig. Residual heatmap of the model given by Eq (5) in S1 Text.**
(TIF)

**S9 Fig.** A is the scatter plot of expected values and the residuals. B shows a histogram of the residuals as well as a blue curve giving the probability density function of a normal distribution

having the same mean and variance as the residuals.
(TIF)

**S10 Fig. Transition probability matrix based on the regrouped scales.**
(TIF)

**S11 Fig. The ratio of the entries in the transition probability matrices for the clusters to the transition probabilities estimated from the whole sample without clustering.**
(TIF)

**S12 Fig. Four clusters without regrouping (Mood, Pain) states.**
(TIF)

**S13 Fig. Heatmaps of transition probability matrices when number of clusters is 5.**
(TIF)

**S14 Fig. Heatmaps of transition probability matrices when number of clusters is 6.**
(TIF)

**S15 Fig. Heatmaps of transition probability matrices when number of clusters is 7.**
(TIF)

**S16 Fig. Heatmaps of transition probability matrices when number of clusters is 8.**
(TIF)

**S1 Table. Conditions reported by the participants of the study.**
(XLSX)

**S2 Table. Sites of chronic pain reported by the participants of the study.**
(XLSX)

**S3 Table. Rounded off values of mean age and response rate.**
(XLSX)

**S4 Table. 8618 out of 9990 participants of the study, reported their chronic pain condition.** Log odds ratio of a condition in a cluster with 95% Confidence Interval.
(XLSX)

**S5 Table. 9146 out of 9990 participants of the study reported their site of pain.** Log odds ratio of a site of pain in a cluster with 95% Confidence Interval.
(XLSX)

**S1 Text. Supplementary text.**
(PDF)

## Author Contributions

**Conceptualization:** Rajenki Das, Mark Muldoon, Mark Lunt, Thomas House.

**Formal analysis:** Rajenki Das.

**Funding acquisition:** Thomas House.

**Investigation:** Rajenki Das, Mark Muldoon, Thomas House.

**Methodology:** Rajenki Das, Mark Muldoon, Mark Lunt, Belay Birlie Yimer, Thomas House.

**Project administration:** Thomas House.

**Resources:** Mark Lunt, John McBeth, Belay Birlie Yimer.

**Software:** Rajenki Das, Mark Muldoon.

**Supervision:** Mark Muldoon, Mark Lunt, Thomas House.

**Validation:** Mark Muldoon.

**Visualization:** Rajenki Das.

**Writing – original draft:** Rajenki Das.

**Writing – review & editing:** Rajenki Das, Mark Muldoon, Mark Lunt, John McBeth, Belay Birlie Yimer, Thomas House.

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
