## [Decision Letter · Decision Letter 0]

18 Nov 2022

PDIG-D-22-00293

Modelling and classifying joint trajectories of self-reported mood and pain in a large cohort study

PLOS Digital Health

Dear Dr. Das,

Thank you for submitting your manuscript to PLOS Digital Health. After careful consideration, we feel that it has merit but does not fully meet PLOS Digital Health's publication criteria as it currently stands. Therefore, we invite you to submit a revised version of the manuscript that addresses the points raised during the review process.

Please submit your revised manuscript within 30 days Dec 18 2022 11:59PM. If you will need more time than this to complete your revisions, please reply to this message or contact the journal office at digitalhealth@plos.org. Please include the following items when submitting your revised manuscript:

We look forward to receiving your revised manuscript.

Kind regards,

Iain Buchan, MD FFPH FACMI

Guest Editor

PLOS Digital Health

Journal Requirements:

a. State what role the funders took in the study. If the funders had no role in your study, please state: “The funders had no role in study design, data collection and analysis, decision to publish, or preparation of the manuscript.”

b. If any authors received a salary from any of your funders, please state which authors and which funders.

2. We ask that a manuscript source file is provided at Revision. Please upload your manuscript file as a .doc, .docx, .rtf or .tex.

Additional Editor Comments (if provided):

Thank you for this well-reasoned paper on pain-mood endophenotype exploration, with methodology relevant to wider mental-physical health comorbidity research.

Please provide the clarifications the reviewers request, mostly in the supplementary material.

Please make your code available in a public repository, and encourage the data-controller to make suitable synthetic data available for easier knowledge-transfer of your methods/code.

Align your clarification of model selection with the code you publish, linked to from the supplementary material.

Clarify the cohort definition/characterisation with more information about https://www.cloudywithachanceofpain.com in the supplement.

Optionally, you may wish to elaborate on the use of (variable) longitudinal observation designs to distinguish patterns of resilience to mood/pain worsening from patterns of susceptibility, with potentially earlier tipping points and different sequencing.

Reviewers' comments:

Reviewer's Responses to Questions

**Comments to the Author**

1. Does this manuscript meet PLOS Digital Health’s publication criteria? Is the manuscript technically sound, and do the data support the conclusions? The manuscript must describe methodologically and ethically rigorous research with conclusions that are appropriately drawn based on the data presented.

Reviewer #1: Yes

Reviewer #2: Yes

2. Has the statistical analysis been performed appropriately and rigorously?

Reviewer #1: Yes

Reviewer #2: Yes

3. Have the authors made all data underlying the findings in their manuscript fully available (please refer to the Data Availability Statement at the start of the manuscript PDF file)?

Reviewer #1: No

Reviewer #2: No

4. Is the manuscript presented in an intelligible fashion and written in standard English?

Reviewer #1: Yes

Reviewer #2: Yes

5. Review Comments to the Author

Reviewer #1: Thank you for the opportunity to review this paper, which I enjoyed and found interesting and informative. 

With specific reference to the PLOS Reviewer Guidelines 

** What are the main claims of the paper and how significant are they for the discipline?

The authors use an existing, longitudinal mHealth data set consisting of > 9000 participants who provide daily, paired (mood,pain) ratings. The authors dichotomise pain and mood into "low" and "high" states to simplify the data to enable an EM fit for mixture Markov models such that for each person, they reside in one of four possible states and can transition to any of these four at the next measurement. 

The authors show how they arrive at a k=4 cluster Markov chain model and provide post-hoc analyses of the distributions of pain conditions within/between the discovered clusters.

Overall, my impression is that this is novel work and methodologically sound (within the confines of my experience with these kinds of modelling techniques). The results are interesting and the limitations discussed appropriately. My interpretation of this paper is that the authors provide a novel method for analysing mHealth data in the context of multi-morbidity in pain syndromes (rather than, say, providing definitive population level analyses with clinical significance). 

These methods would be applicable and have utility for researchers in other conditions/disorders where understanding paired associations over time would provide insights. For example, as the authors show, chronic headache and fibromyalgia (and to some extent, spondyloarthropathy) have higher log odds of being present in Cluster 2 participants and this represents an insight that mood/affective state modulates experiences of pain (or vice versa) in these conditions in a way that separates them from participants in other clusters.

I have only minor suggestions, outlined below, which I think would benefit the paper.

** Are these claims novel? If not, please specify papers that weaken the claims of originality of this one. If a similar paper was recently published, the current manuscript may still be eligible for publication under our complementary research policy?

The research is novel.

** Are the claims properly placed in the context of the previous literature? Have the authors treated the literature fairly?

Yes. 

** Do the data and analyses fully support the claims? If not, what other evidence is required?

Yes.

** Would additional work improve the paper? How much better would the paper be if this work were performed and how difficult would it be to do this work?

I would suggest the following minor amendments

 1) the authors don't provide detail on whether or not the participants were self-selecting and self-reporting diagnoses. This would help situate the paper for clinically-focused readers who might ask whether or not the participants were a clinical population (i.e. any diagnoses of mood disorders and/or confirmed diagnoses of pain syndromes / musculo-skeletal / rheumatoid disorders). The introduction helpfully describes the pain <-> mood bidirectional association, so it would be worth highlighting if this is a "clinical population" or a group of people self-selecting to trial monitoring their mood <-> pain symptoms. Alternatively, in the Limitations sections, the authors could clarify if this can be considered a clinical population or not. and what is known about participants' diagnoses.

 2) I struggled to understand how the "intervention" (which appears to simulate changes in individual cluster's transition probability matrices) simulations were implemented -- for example, after fitting the k = 4 model, from my reading, it seems the authors then set the beta_P and beta_M values (in addition to the transition probability matrix contraints -- constants 0.8 and 0.2 -- to impose an asymmetric transition probability BH and BL from BG and BL) so that beta_M < 0.15 and beta_P < 0.2. If the intention is to show that the model could allow one to simulate intervention, then I would either expand on this in the main text, or if this is less important, I would move to supplementary information -- as it stands, I couldn't find an a priori reason why this simulation was included in the main manuscript (especially given they chose the constants/penalties and beta values themselves). I think I understand the what the authors are attempting to show, but my overall impression is that this is somewhat arbitrary and I'm not entirely clear that it adds much to the results presented. 

 3) Model selection: The authors argue for a qualitative "elbow" in the negative log-likelihood implying a solution with clusters k = 4 (Figure S4, section S.1.4) and that "4-component mixture had certain natural interpretations". While this is reasonable, I wondered why the authors did not select clusters using a method that explicitly penalises for model complexity (i.e. the number of parameters) such as BIC ? If the data supports a k = 4 cluster model, one would expect the resulting penalised neg log lik to be parabolic, with a minimum at 4 -- if this is the best compromise between under/over-fitting the number of clusters ? Perhaps the authors could comment on this in Suppl Info. 

** If a protocol is provided, for example for a randomized controlled trial, are there any important deviations from it? If so, have the authors explained adequately why the deviations occurred?

Not relevant to this work.

** PLOS Digital Health encourages authors to publish detailed protocols and algorithms as supporting information online. Do any particular methods used in the manuscript warrant such treatment?

I defer to the editor on this issue, but my view is that for methods such as those presented, the authors should place their code in a public repository so others can inspect the finer-grained implementation of the methods. No libraries/implementation language is mentioned in the text, which suggests the authors implemented the methods themselves, highlighting the need for transparent code. The data is highlighted as being owned by a party that is not the authors, so I would encourage them to provide some minimal, simulated example data so others can explore their code/implementation without needing to locate similar data. 

** If the paper is considered unsuitable for publication in its present form, does the study itself show sufficient potential that the authors should be encouraged to resubmit a revised version?

Not relevant

** Are original data deposited in appropriate repositories and accession/version numbers provided for genes, proteins, mutants, diseases, etc.?

See above : the authors indicate they are unable to provide the data and on the cited project website (https://www.cloudywithachanceofpain.com/blog), there is no data sharing statement that I could locate.

** Does the study conform to any relevant guidelines such as CONSORT, MIAME, QUORUM, STROBE, and the Fort Lauderdale agreement?

Not relevant.

** Are details of the methodology sufficient to allow the experiments to be reproduced?

Partially, see above. 

** Is any software created by the authors freely available?

See above.

** Is the manuscript well organized and written clearly enough to be accessible to non-specialists?

Yes.

Reviewer #2: This is a nicely written study aiming to identify patterns of pain and mood reporting in a large electronic health dataset. They show that although pain and mood are known to be associated, there are groups of people who have different relationships between pain and mood, showing 4 distinct categories. The approach is appropriate and interesting, and the findings may have value in future targeting of personalised treatments for pain and low mood.

I only have a few minor comments and requests for clarification.

1. For how many days were participants expected to respond on the App? What happened if a person missed one day but then continued to fill in the app?

2. How were the odds ratios in tables S2 and S3 calculated? Something like a multinomial model? Can this be clarified in the methods section please?

3. The choice has been made to identify 4 clusters. However, in Figure S4, more clusters provide better log-likelihood. The justification is that the reduction is not much once you have more than 4 clusters. But this is a bit arbitrary. It would be good to discuss this a little more and describe what impact this might have. How strongly should we believe that there are four patterns of behaviour for example? In say the 8 or 10 cluster model, do you get lots of similar clusters, or do you find additional patterns to the ones you have identified? The abstract says that you have discovered four distinct patterns – how reliable is this?

4. What impact do you think reducing the states to 2 per variable has? It’s certainly simpler, but do you think it loses useful information on trajectory?

5. How reasonable is the Markov assumption in equation 3? I wonder the next state being only dependent on the previous state (and not how it got there) is unlikely to hold – especially in the 25-state model. I suspect this may be less of an issue in the collapsed model presented here with only 2 categories for each of pain and mood. But in the 5 original categories might it not be the case that someone in state 3 was more likely to go to state 4 if they had shown a constant rise from state 1 through to state three than if they had just arrived in state 3 from state 4?

6. The clustering of Markov processes described in this paper is a neat idea. Setting this problem up as a set of states and looking at transitions between them is certainly a natural way to think. I do find clustering of transition matrices a little hard to conceptualise though. There are other ways of doing longitudinal clustering that might work in this setting. For example, latent class trajectory models (Proust-Lima, C., Philipps, V. and Liquet, B., 2015. Estimation of extended mixed models using latent classes and latent processes: the R package lcmm. arXiv preprint arXiv:1503.00890.) or multivariate mixture models (Komárek, A. and Komárková, L., 2013. Clustering for multivariate continuous and discrete longitudinal data. The Annals of Applied Statistics, 7(1), pp.177-200.). I think these would be suitable for your problem (specially with binary pain and mood scores). I’m not proposing redoing any analysis, but I think it would be useful to describe the relative methods of your approach compared to these other examples of approaches to doing essentially the same thing.

6. PLOS authors have the option to publish the peer review history of their article (what does this mean?). If published, this will include your full peer review and any attached files.

**Do you want your identity to be public for this peer review?** For information about this choice, including consent withdrawal, please see our Privacy Policy.

Reviewer #1: No

Reviewer #2: No

---

## [Editor Report · Decision Letter 1]

27 Jan 2023

Modelling and classifying joint trajectories of self-reported mood and pain in a large cohort study

PDIG-D-22-00293R1

Dear Miss Das,

We are pleased to inform you that your manuscript 'Modelling and classifying joint trajectories of self-reported mood and pain in a large cohort study' has been provisionally accepted for publication in PLOS Digital Health.

Best regards,

Iain Buchan, MD FFPH FACMI

Guest Editor

PLOS Digital Health

Thank you for your revisions, which address reviewers' comments and highlight the need for methodological development with temporally and causally complex symptom/experiential data and contextual metadata.